EMBO
Molecular Medicine

# Rescuing ocular development in an anophthalmic pig by blastocyst complementation

Hongyong Zhang[1,2,3,†], Jiaojiao Huang[1,2,3,†], Zechen Li[4,†], Guosong Qin[1,2,3], Nan Zhang[1], Tang Hai[1,2,3], Qianlong Hong[1], Qiantao Zheng[1,2,3], Ying Zhang[1,2,3], Ruigao Song[1,2,3], Jing Yao[1,2,3], Chunwei Cao[1,2,3], Jianguo Zhao[1,2,3,*] (iD) & Qi Zhou[1,2,3,**] (iD)

## Abstract

Porcine-derived xenogeneic sources for transplantation are a promising alternative strategy for providing organs for treatment of end-stage organ failure in human patients because of the shortage of human donor organs. The recently developed blastocyst or pluripotent stem cell (PSC) complementation strategy opens a new route for regenerating allogenic organs in miniature pigs. Since the eye is a complicated organ with highly specialized constituent tissues derived from different primordial cell lineages, the development of an intact eye from allogenic cells is a challenging task. Here, combining somatic cell nuclear transfer technology (SCNT) and an anophthalmic pig model (*MITF*[L247S/L247S]), allogenic retinal pigmented epithelium cells (RPEs) were retrieved from an E60 chimeric fetus using blastocyst complementation. Furthermore, all structures were successfully regenerated in the intact eye from the injected donor blastomeres. These results clearly demonstrate that not only differentiated functional somatic cells but also a disabled organ with highly specialized constituent tissues can be generated from exogenous blastomeres when delivered to pig embryos with an empty organ niche. This system may also provide novel insights into ocular organogenesis.

**Keywords** anophthalmic pig; chimera; organ regeneration; somatic cell nuclear transfer technology
**Subject Categories** Development & Differentiation; Genetics, Gene Therapy & Genetic Disease; Regenerative Medicine

## Introduction

Organ transplantation is the only therapy solution for those patients suffering from end-stage organ failure and fatal diseases (Denner, 2017). The shortage of donor organs has limited the number of patients that can obtain treatment, and many individuals die before the organs become available. Thus, the generation of transplantable organs is one important goal of stem cell-based regenerative medicine (Rami *et al*, 2017). Pluripotent stem cells (PSCs) have opened new avenues for the treatment of degenerative diseases using patient-specific stem cells to generate tissues and cells (Shi *et al*, 2017). However, the generation of a functional organ from PSCs has not been feasible because it remains difficult to mimic *in vitro* the sophisticated interactions among cells and tissues during organogenesis (Kobayashi & Nakauchi, 2011). This difficulty might be addressed by generating organs *in vivo* using a blastocyst complementation strategy, which was first reported by Chen *et al* (1993) to generate mature B and T lymphocytes. Recently, two studies from the Nakauchi laboratory have reported proof-of-principle findings to demonstrate that functional organs—kidney and pancreas—could be generated from PSCs *in vivo* using blastocyst complementation in organogenesis-disabled mouse embryos (Kobayashi *et al*, 2010; Usui *et al*, 2012). Furthermore, with rat iPS cell injection, blastocyst complementation successfully rescued pancreas, hearts, and eyes in organogenesis-disabled mouse offspring (Wu *et al*, 2017). Considering ethical issues, infectious disease concerns, and physiological characteristics and size, larger mammals are preferred for generating transferable human organs, and pigs are the most suitable choice (Niemann & Petersen, 2016). Studies from Matsunari *et al* (2013) confirmed that the blastocyst complementation strategy is feasible in a large-animal model, using apancreatic pigs to generate a functional pancreas with allogenic blastomeres.

The eye is a complicated organ with highly specialized constituent tissues derived from different primordial cell lineages (Hayashi *et al*, 2016). Age-related ocular degenerative diseases, such as

---

1   State Key Laboratory of Stem Cell and Reproductive Biology, Institute of Zoology, Chinese Academy of Sciences, Beijing, China
2   Savaid Medical School, University of Chinese Academy of Sciences, Beijing, China
3   Institute of Stem Cell and Regeneration, Chinese Academy of Sciences, Beijing, China
4   College of Life Sciences, Qufu Normal University, Qufu, China
    *Corresponding author. Tel: +86 10 64806259; E-mail: zhaojg@ioz.ac.cn
    **Corresponding author. Tel: +86 10 64806299; E-mail: qzhou@ioz.ac.cn
    †These authors contributed equally to this work

---

retinal degeneration or age-related macular degeneration (AMD) and retinitis pigmentosa, are difficult to cure and are characterized by the dysfunction and death of light-sensitive photoreceptors and RPE (Forest *et al*, 2015; Mellough *et al*, 2015). The diseases have shown significant promise in being treated with PSCs. Stem cell-based therapies, such as subretinal injection of human embryonic stem cell (hESC)-derived RPE cells, adult autologous induced pluripotent stem cell (iPSC)-RPE, neural stem cells, umbilical cord stem cells, or bone marrow stem cells, have been the most important tools used to treat AMD (Eiraku *et al*, 2011; Nakano *et al*, 2012; Reichman *et al*, 2014; Zhong *et al*, 2014; Forest *et al*, 2015; Mellough *et al*, 2015; Song *et al*, 2015). Several studies have reported that PSCs can be induced to differentiate along a retinal lineage, including differentiation into photoreceptors using specifically defined culture conditions (Ikeda *et al*, 2005; Lamba *et al*, 2006; Osakada *et al*, 2008; Meyer *et al*, 2009; Boucherie *et al*, 2013). Moreover, a three-dimensional optic cup can be formed *in vitro* from mouse or human embryonic stem cells and can develop into a structure that remarkably resembles the embryonic vertebrate eye (Nakano *et al*, 2012; Boucherie *et al*, 2013). However, recently two studies reported less compelling results in patients with respect to stem cell-based therapy for AMD (Kuriyan *et al*, 2017; Mandai *et al*, 2017) and Kuriyan *et al* (2017) reported that three patients encountered severe bilateral visual loss that developed after they received intravitreal injections of autologous adipose tissue-derived "stem cells" at a private clinic in the United States. Other clinical trials also failed to show functional improvements in macular degeneration patients, possibly because of immune rejection and graft failure (Kimbrel & Lanza, 2015; Song *et al*, 2015). The failure of PSC-based therapy suggests that *in vitro* culture system cannot mimic the *in vivo* environment completely and it is unclear to what extent hPSCs can recapitulate the cellular and molecular features of native RPE *in vitro*. Thus, our results suggest that high-quality characterized RPE cells from *in vivo* differentiation systems or intact eyes might provide alternative solutions to address the safety and technical challenges of stem cell-based therapies for ocular degenerative diseases.

In the current study, we demonstrate that intact eyes can be regenerated from allogenic blastomeres *in vivo* using complementation of organogenesis-disabled pig embryos. The regenerated eyes in the chimeric pig show normal configuration and function. In addition, allogenic-characterized RPEs can be generated from E60 fetuses, which enable the organ-defective fetus to be a niche for *in vivo* differentiation. Blastocyst complementation, using somatic cloned, organ-defective pig embryos, may thus permit the use of a large animal to generate functional and complex organs such as eyes from xenogenic PSCs.

# Results

### Generation of porcine chimeric embryos *in vitro* by blastocyst complementation

To generate allogenic chimeric pigs, we first explored the possibility of blastocyst complementation *in vitro* using cloned embryos derived from pig embryonic fibroblast cells (PEFs; Fig EV1A). PEFs derived from Bama miniature pigs were labeled with either red

fluorescence protein (RFP) or green fluorescence protein (GFP) and then used as donors for SCNT (Fig EV1B). Somatic cloned embryos derived from RFP-positive PEFs at the 4-cell or 8-cell stage (day 3) were used as host embryos, and ~ 5 GFP-labeled blastomeres (day 4) were injected as donors for the generation of the chimeric embryos (Fig EV1C). The reconstructed embryos were further cultured for 3–4 days and then assessed for blastocyst formation and genotyping (Fig EV1D). The injection of donor blastomeres did not affect the developmental competency of reconstructed embryos as evidenced by the similar blastocyst rates between complemented embryos and non-injected SCNT embryos ($23.47\% \pm 1.685$ vs. $18.57\% \pm 1.434$, $P = 0.09$; Table 1). In the observed blastocysts, 9 of 10 (90%) expressed both *RFP* and *GFP*, indicating successful chimerism *in vitro* (Fig EV2A). To further confirm the feasibility of blastocyst complementation in PEFs with a different genetic background, somatic cloned embryos derived from PEFs that carried a lysine-to-serine substitution (L247S) in the microphthalmia-associated transcription factor (*MITF*) (termed as *MITF*^L247S/L247S) were injected with Large White (LW) GFP-positive blastomeres. Our results showed that most injected *MITF*^L247S/L247S embryos expressed *GFP* during blastocyst formation, consistent with our previous findings (Fig EV2B). Restriction fragment length polymorphism (RFLP) analysis with single blastocyst PCR amplification was used to characterize the *MITF*^L247S/L247S mutation and substantiate the chimerism. Results showed that 7 of 10 blastocysts were chimeric (Figs EV2C and EV3A). The findings confirm that blastocyst complementation is feasible *in vitro* with embryos derived from PEFs.

### The anophthalmic phenotype was repaired in E44 *MITF*^L247S/L247S porcine fetus

*MITF*^L247S/L247S confers an eye developmental defect and can be observed as early as embryonic day 28 (E28; Hai *et al*, 2017b). To further investigate the developmental competency of pig embryos derived from the blastocyst complementation, cloned embryos derived from *MITF*^L247S/L247S male PEFs were used as the host and blastomeres of cloned embryos derived from female LW PEFs were used as donors for complementation. Reconstructed embryos at day 5 were transferred to surrogates for further development evaluation. A total of 3,754 embryos were transferred to 16 surrogate sows, and three became pregnant (Appendix Table S2). Of the three pregnancies, two terminated during early pregnancy and four fetuses (named NW-1, NW-2, NW-3, and NW-4) were retrieved at E44 by Caesarean section from one litter (Fig 1A). The four fetuses demonstrated normal shape and size compared with control wild-type (WT) and *MITF*^L247S/L247S cohorts produced by natural mating (Fig 1A). Genotyping results showed that the NW-2 fetus was chimeric in all the tissues tested, whereas the NW-4 fetus was derived from donor embryos and the NW-1 and NW-3 fetuses were derived from host embryos (Figs 1B and EV3B). The chimerism of NW-2 was further confirmed by the characterization of an LW-specific *KIT* allele. A *KIT* duplication is seen in LW pigs but not in Bama pigs, so a pair of primers was designed to generate a 152-bp PCR product spanning the 3′–5′ breakpoint as a diagnostic test for the duplication of *KIT* in the chimeric fetus (Giuffra *et al*, 2002). Results further confirmed that the LW-specific fragment could be detected in the NW-2 and NW-4 fetuses but not in the NW-1 and NW-3 fetuses (Fig 1B).

**Table 1. The blastocyst rate between the SCNT embryo and complementation embryo.**

| Treatment | No. of replications | No. of embryos | No. of blastocysts (%) | P-value |
|---|---|---|---|---|
| Non-injected SCNT embryo | 3 | 371 | 69 (18.57 ± 1.434)[a] | |
| Complementation embryo | 3 | 380 | 89 (23.47 ± 1.685)[a] | 0.09 |

[a]t-test indicates that values are not significantly different (P = 0.09), the results are presented as means ± SEM. A P-value of < 0.05 was considered statistically significant.

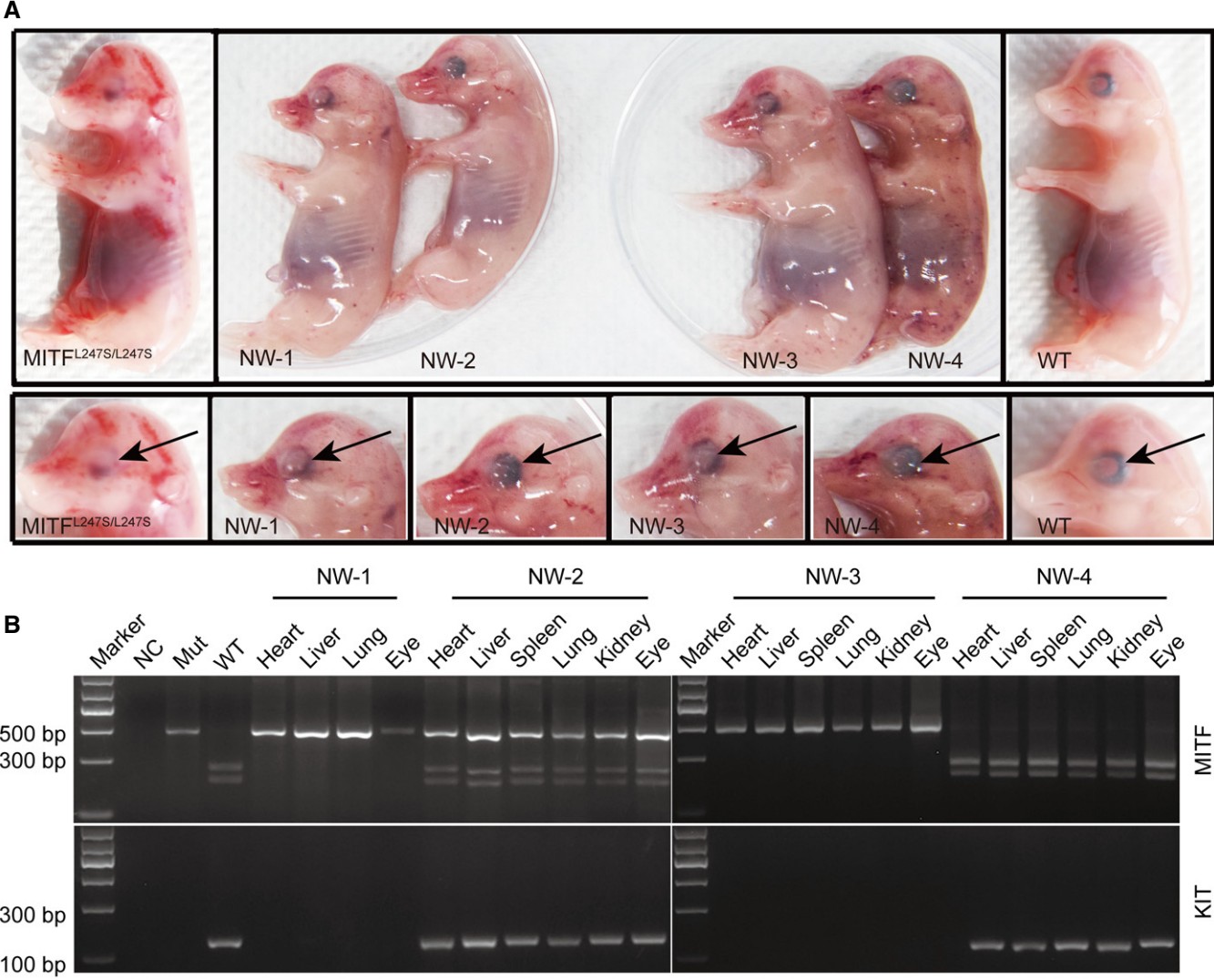

**Figure 1. Generation of E44 chimeric porcine fetus *in vivo* by complementation of *MITF*[L247S/L247S] embryos with donor blastomeres derived from LW PEFs.**

A Four fetuses (named NW-1, NW-2, NW-3, and NW-4) with different eye morphology were retrieved at E44 by Caesarean section. The arrows identify the melanin in the eye.

B DraI digestion of PCR products from the multiple organs of the four fetuses (top). NC, negative control with no genomic DNA loaded. Mut, *MITF*[L247S/L247S] genomic DNA loaded. WT, LW genomic DNA loaded. Multiple organs of the four fetuses were examined for *KIT* expression by agarose gel electrophoresis (bottom). Mut, *MITF*[L247S/L247S] genomic DNA loaded. WT, LW genomic DNA loaded.

Source data are available online for this figure.

The *MITF*[L247S/L247S] E44 fetus displayed an anophthalmic phenotype and showed a loss of RPE cells (Steingrimsson *et al*, 2004; Hai *et al*, 2017b; Fig 1A). The eyes of NW-2 showed normal morphologically and were similar to those of the wild-type (WT) fetus (Fig 1A). Hematoxylin and eosin (H&E) staining showed that the RPE cells of NW-2 were normal and pigmented, whereas the RPEs of *MITF*[L247S/L247S] fetus were hypopigmented and disorganized (Fig 2A). Moreover, positive staining of RPE-specific markers, *MITF*, *Pax6*, and *Bestrophin* (Song *et al*, 2015), were observed in the regenerated RPE of NW-2 fetus, suggesting characterized RPE in NW-2 (Fig 2B). Interestingly, we found that *MITF*[L247S/L247S] mutant showed disordered subcellular distribution of *Pax6* and *MITF* and

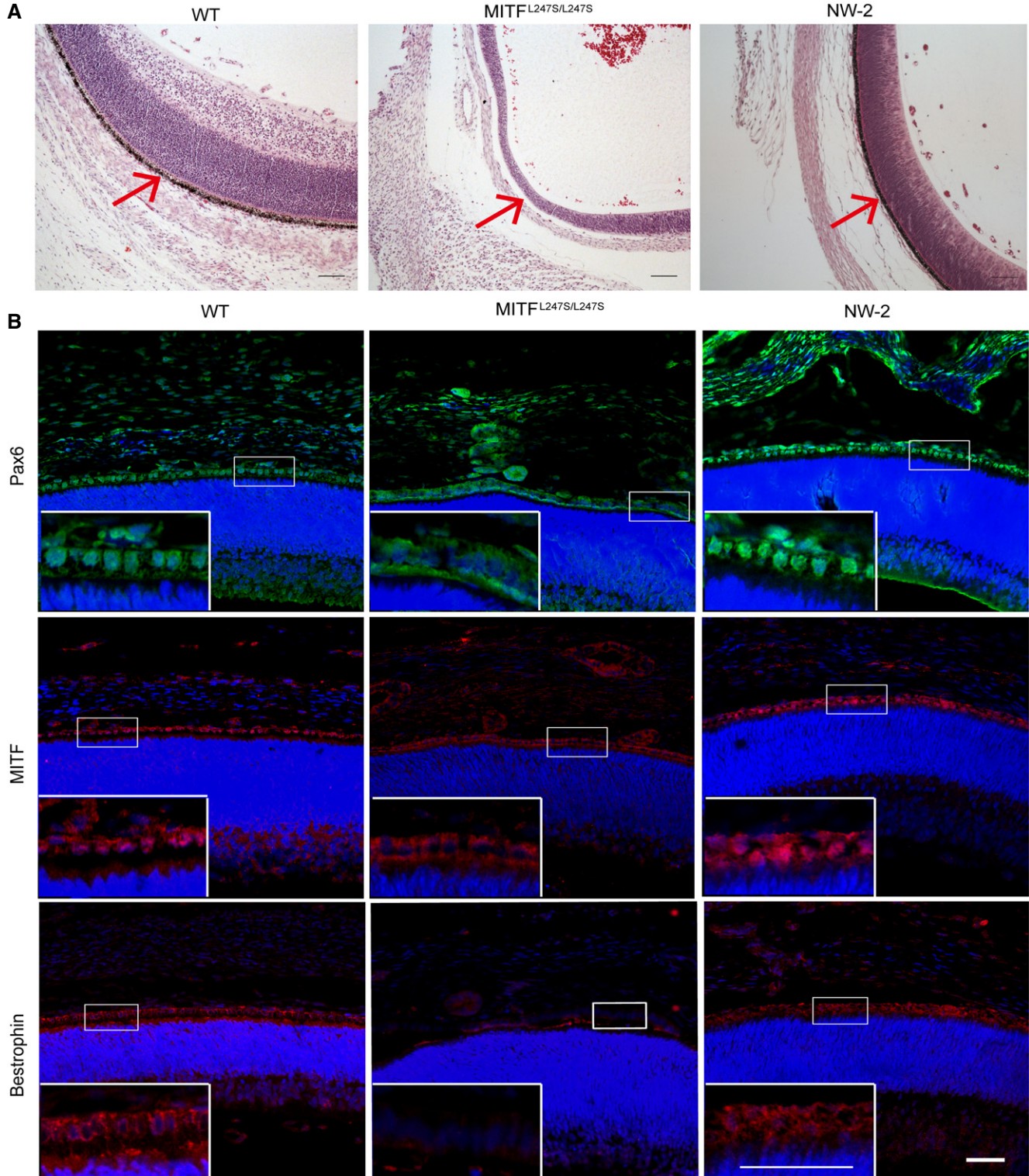

**Figure 2. Allogenic contribution and rescue RPEs in the E44 chimeric fetus.**

A   Representative microscopic appearances of the retina and RPE cells (arrowhead) of a chimeric, a WT, and a Mut fetus at the same gestational age by H&E staining. WT, Bama WT fetus. Mut, Bama $MITF^{L247S/L247S}$ fetus. NW-2, the chimeric fetus. Scale bars, 100 μm.

B   Representative immunofluorescence images showed the expression of multiple RPE markers *Pax6*, *MITF*, and *Bestrophin* in the chimeric fetuses. Blue, Hoechst 33342. WT, Bama WT fetus as positive control. NW-2, the chimeric fetus. $MITF^{L247S/L247S}$ fetus as negative control. Scale bars, 50 μm.

Source data are available online for this figure.

negative expression of *Bestrophin*. In the Hai *et al* manuscript, they also found that the *MITF* mutation affected the subcellular distribution of MITF protein. Finally, using next-generation sequencing of liver, lung, kidney, and eye, we examined the chimerism

contributions and showed the percentage of chimerism was 27.85, 28.06, 27.12, and 27.14%, respectively (Fig EV4A). Our data demonstrate that using blastocyst complementation, allogenic eyes with RPE cells can be regenerated in pigs at E44.

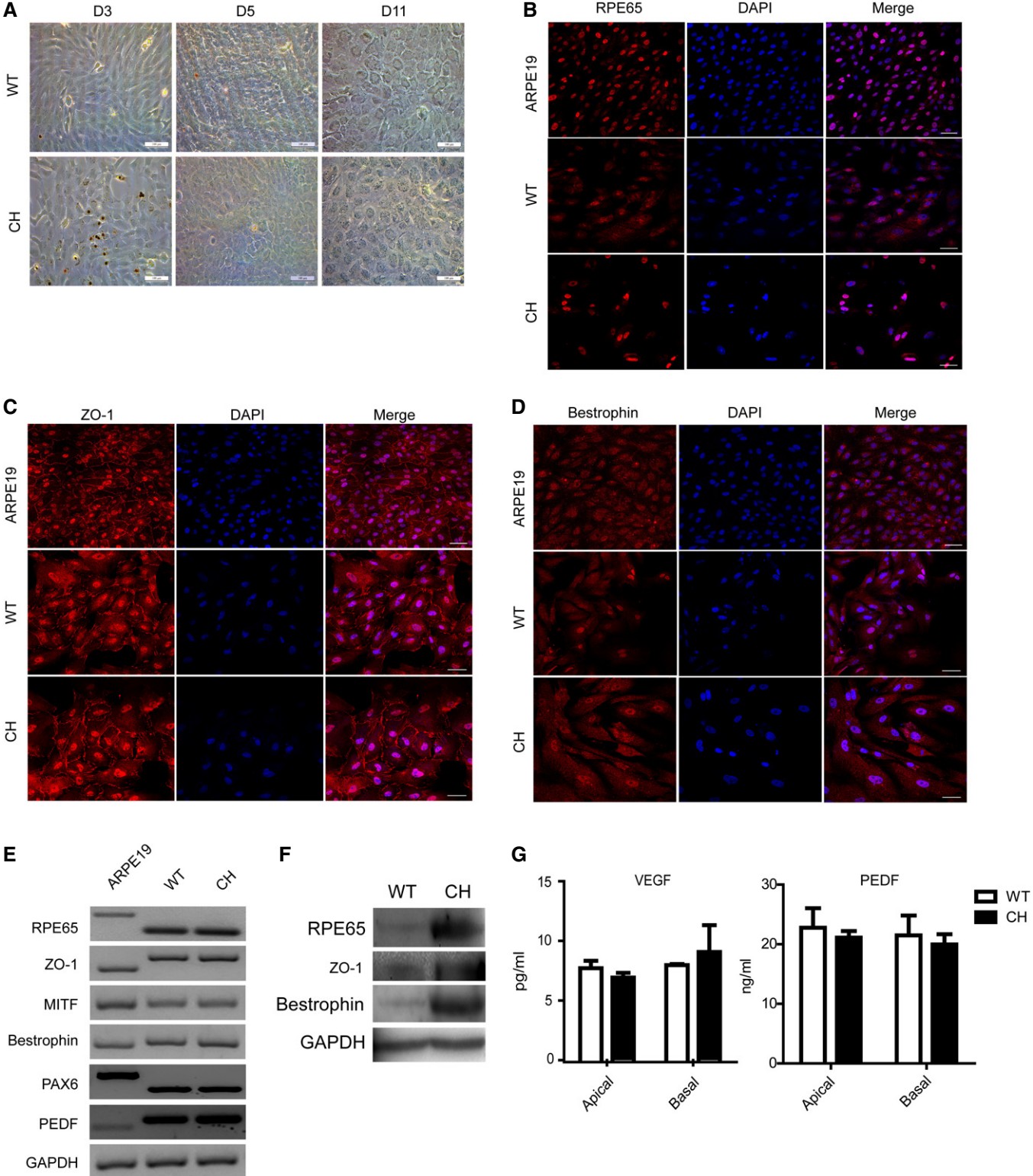

Figure 3.

## RPE cells isolated and characterized from the E60 chimeric fetus

It had been previously thought that functional human RPEs could be differentiated from either PSCs (Liao *et al*, 2010; Fields *et al*, 2016; Sugita *et al*, 2016) or cultured human fetal RPE cells directly *in vitro* (Maminishkis *et al*, 2006; Sonoda *et al*, 2009) for cell therapy. However, the *in vitro* differentiation of PSCs into RPEs still needs to be optimized to address safety and effectiveness concern, and human fetal RPE cells are still limiting, due to a lack of donor resources. Thus, we explored whether allogenic RPE cells could be isolated from chimeric fetuses. The allogenic fetuses constructed from $MITF^{L247S/L247S}$ PEF-derived embryos injected with Bama WT blastomeres were collected at E60 and used to generate RPE cells. At this stage, RPE cells are formed, mature, and easily peeled away from the choroid. A total of 821 embryos were transferred to three surrogate sows, and one sow became pregnant. In one pregnancy, seven fetuses were retrieved and one fetus (NW-7, 1/7, 14.29%) was proved to be chimeric. The other five fetuses (NW-5, NW-6, NW-9, NW-10, NW-11) were derived from host embryos, and 1 (NW-8) was derived from donor embryos (Fig EV3C). The contribution of the donor cells was identified in various tissues in the chimeric fetus (NW-7), for which the NW-8 fetus was used as control (Fig EV3D). RPE cells were isolated from NW-7 and NW-8 and characterized as successfully hexagonal mosaic (Fig 3A). RPE-specific markers of *RPE65* (Fig 3B), *ZO-1* (Fig 3C), and *Bestrophin* (Fig 3D) were clearly identified using immunofluorescence staining, and human ARPE19 cells were compared as a positive control. Semi-quantitative RT–PCR analyses and Western blotting further confirmed the characteristics of RPEs (Fig 3E and F). VEGF and PEDF angiogenic factors, which are secreted specifically by RPE cells (Maminishkis *et al*, 2006), were also examined by ELISA assay in NW-7 and NW-8 RPE cells. Assay results showed trends that VEGF was secreted more into the basal bath ($n = 3$, $P = 0.4737$), whereas PEDF was secreted more into the apical bath ($n = 3$, $P = 0.2435$) in CH RPE cells, but overall, there was no significant difference (Fig 3G). Together, these results demonstrate that allogenic porcine RPE cells could be isolated from E60 fetuses and characterized *in vitro*.

## Full-term chimeras with enriched contribution of allogenic cells to the defective organ niche

To investigate whether the eyes could be regenerated in the $MITF^{L247S/L247S}$ offspring by blastocyst complementation, host embryos derived from Bama $MITF^{L247S/L247S}$ male PEFs were injected with blastomeres derived from GFP-labeled Bama male PEFs. A total of 1,671 reconstructed embryos were transferred into seven surrogate sows, resulting in three pregnancies, two of which developed to term (Appendix Table S3). In two litters, seven live-born piglets were obtained. Of the littermates, one piglet (NW-16) was identified as a chimera, and the others were all cloned piglets derived from $MITF^{L247S/L247S}$ host embryos (Figs 4A and EV3E). The chimeric piglet with intact eyes showed normal size and morphology, suggesting that the contribution of allogenic blastomeres does not affect pig development (Fig 4A). To examine the chimeric contribution in the NW-16 piglet, *MITF* and *GFP* were genotyped in various organs by RFLP or PCR, respectively. In all the organs, we detected *MITF* chimerism and *GFP* presence, as was seen in the E44 chimeric fetus (Figs 4B and EV3F). The structures of the corneas and retinas were similar to those of WT as determined by H&E analysis (Fig 4C and D). Almost all the RPE cells and corneal epithelial cells of the chimeric piglet were *GFP* positive, indicating the origin and enrichment of the donor cells (Fig 5A and B). The percentage of chimerism was 85.76% (6,812/7,943) in the eye of the chimeric piglet as determined by next-generation sequencing analysis (Fig EV4A). A higher chimerism percentage was observed in the eyes than in other organs in the NW-16 piglets (Fig EV4A). Simultaneously, we confirmed that the *GFP*-positive cells expressed the functional markers of retina and corneal epithelial cells (Fig 5C and D). Furthermore, the expression of the *GFP* gene in the testis, lung, and kidney of the chimeric pig was also detected (Fig EV4B). Taken together, our data demonstrate the feasibility of a full rescue of complex organs, such as the eyes, in organ-defective embryos using blastocyst complementation.

## Discussion

Although blastocyst complementation was established to study the development of a number of tissues and organs (Kobayashi *et al*, 2010; Usui *et al*, 2012) in rodents, only one study so far has reported the generation of organs—pancreatic islets—in pigs using this strategy (Matsunari *et al*, 2013). In the current study, we extended this concept further to show that a more sophisticated organ, such as eyes derived from allogenic pluripotent stem cells, can be generated when organogenesis-disabled embryos are complemented with allogenic blastomeres. This was achieved not only in the chimeric fetuses at E44 and E60 but also in the full-term chimeric pig with intact eyes.

The homozygote offspring $MITF^{L247S/L247S}$ were identified from a large-scale ENU mutagenesis program (Hai *et al*, 2017a) and

---

◄ **Figure 3. Characterized porcine RPE cells derived from E60 chimeric fetus by complementation of Bama $MITF^{L247S/L247S}$ embryos with WT blastomeres.**

A    Representative light microscopic images show that the *in vitro*-cultured RPE cells exhibited a hexagonal shape. D3, D5, and D11, cultured for 3, 5, and 11 days, respectively. Scale bars, 100 μm.

B–D  Representative immunofluorescence images showed the expression of RPE-specific markers, including *RPE65* (B), *ZO-1* (C), and *Bestrophin* (D), respectively, in the cultured RPE cells. Scale bars, 50 μm.

E    Semi-quantitative RT–PCR analysis of RPE functional markers in cultured porcine RPE cells. *GAPDH* was used to confirm the cDNA quality of all the samples.

F    Western blotting identified three RPE markers in porcine RPE cells. *GAPDH* served as a housekeeping gene control.

G    ELISA identified secreted VEGF and PEDF in chimeric and WT porcine RPE cells. Media were collected at 48 h after the cells were seeded onto 12-well cell culture plates to be used for analyses. The results are representative of three independent experiments and were evaluated using a two-tailed *t*-test. No significant differences were identified. Data are presented as the mean ± standard error mean (SEM). WT, RPE cells cultured from the NW-8 fetus. CH, RPE cells cultured from chimeric NW-7 fetus.

Source data are available online for this figure.

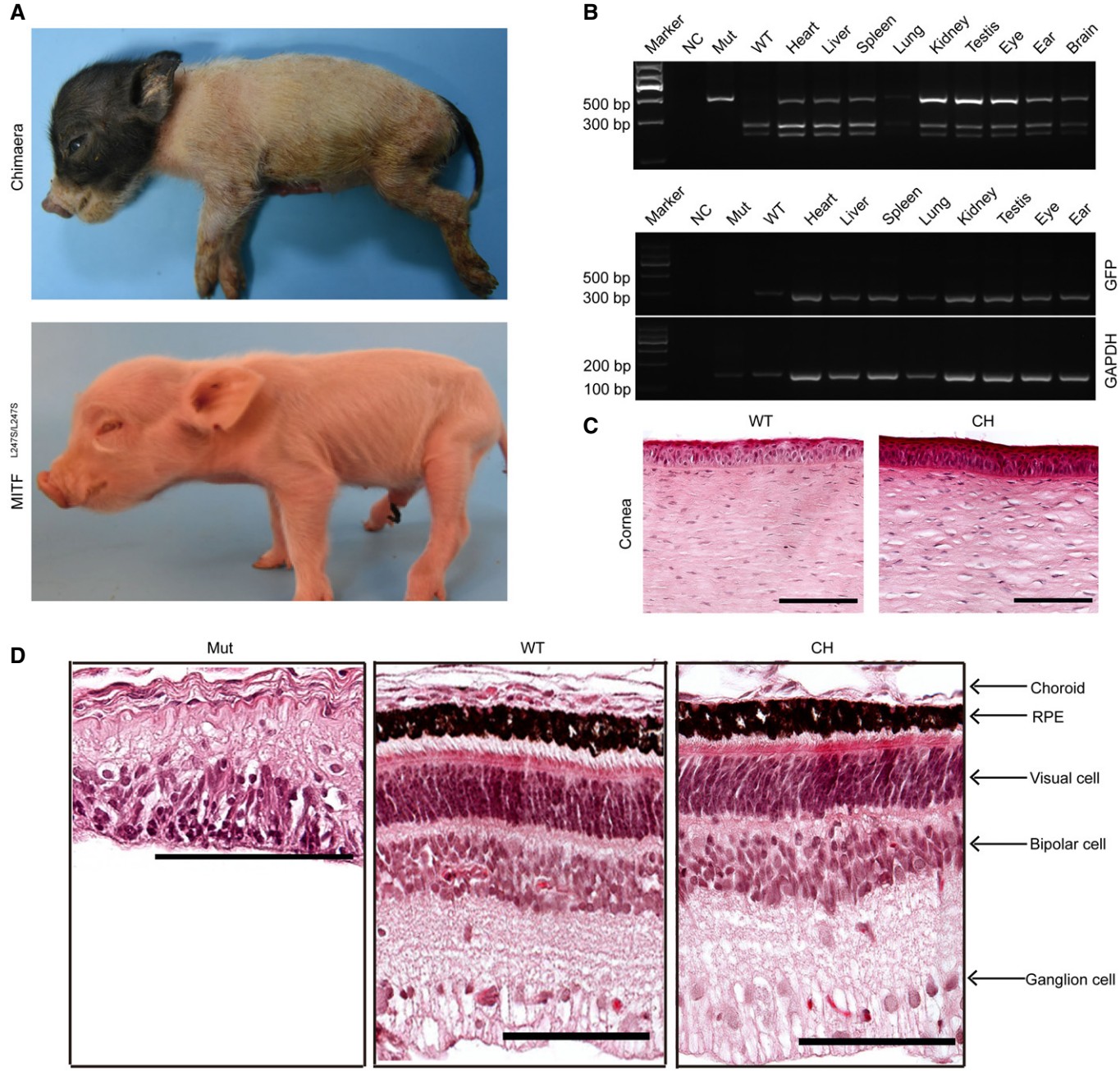

**Figure 4. Generation of a full-term chimeric piglet from Bama *MITF*<sup>L247S/L247S</sup> cloned and Bama GFP-labeled cloned embryos.**

A   The full-term chimera (NW-16) piglet was born and showed normal development, size, and morphology of the eyes.

B   Digestion of the *MITF* PCR products in multiple organs of the full-term chimeric pig. NC, negative control with no genomic DNA loaded. Mut, *MITF*<sup>L247S/L247S</sup> genomic DNA loaded. WT, Bama GFP-labeled genomic DNA loaded. In the chimera piglet, *GFP*-specific primers were used to further confirm the chimeric contribution to multiple organs in the piglet. *GAPDH* was used to confirm the DNA quality of all the samples. NC, negative control with no genomic DNA loaded. Mut, Bama *MITF*<sup>L247S/L247S</sup> genomic DNA loaded. WT, Bama GFP-labeled genomic DNA loaded.

C   Representative microscopic appearances of the corneal epithelial cells of the chimeric piglet staining by H&E. Scale bars, 100 μm.

D   Representative microscopic appearances of the retinal cells of the chimeric piglet staining by H&E. Scale bars, 100 μm.

Source data are available online for this figure.

developed defects in the eyeballs (Hai *et al*, 2017b). This valuable anophthalmic pig model provided us a unique host with an empty developmental niche for exploring the possibility of producing an intact eye from xenogenic PSCs or an *in vivo* differentiation system.

The empty "organ niche" in organogenesis-disabled animals can provide a special microenvironment that not only determines the stem cell fate and differentiation of donor PSCs but also extends to the generation of organs by blastocyst complementation (Miyamoto

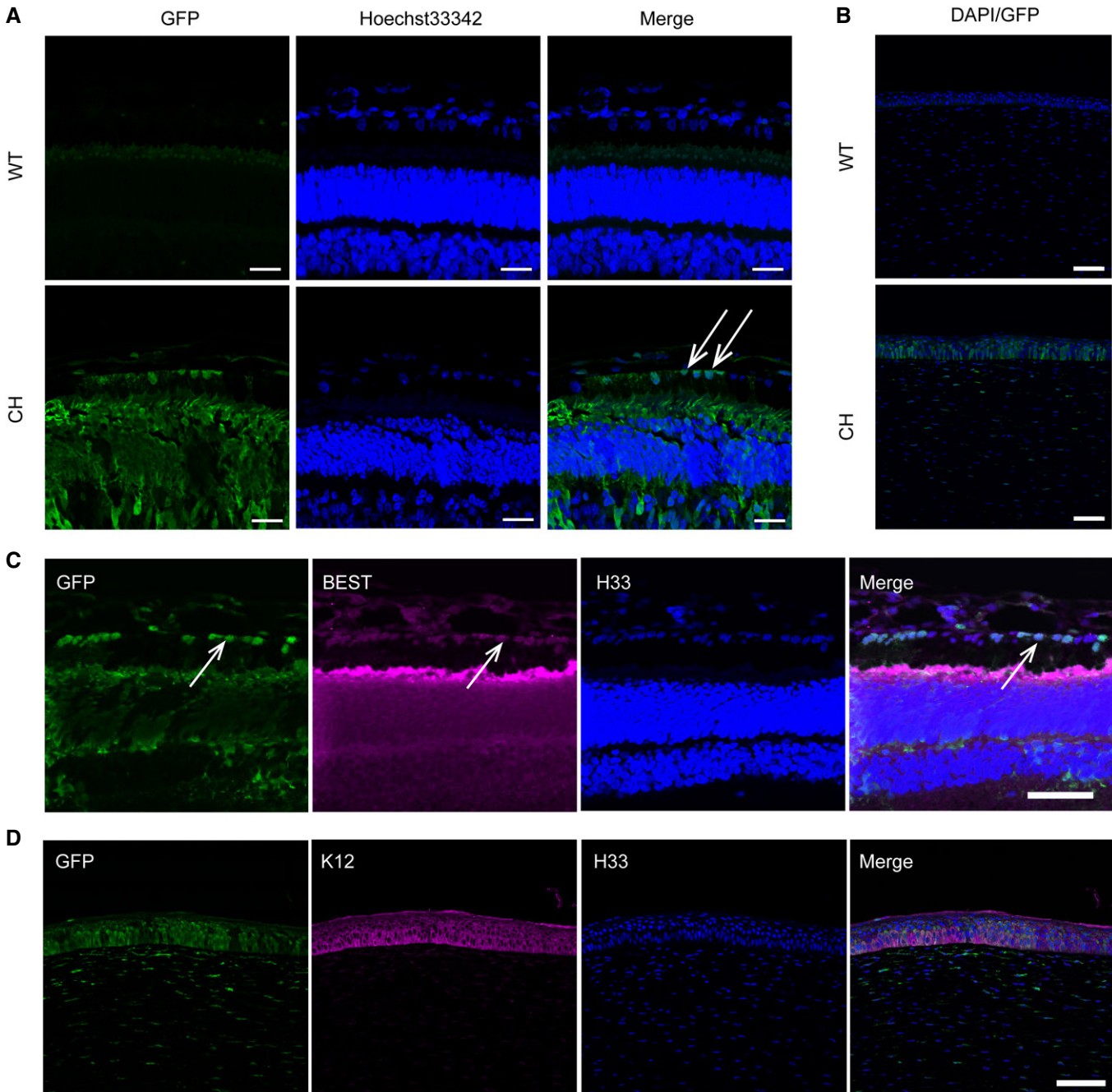

**Figure 5.    The RPEs and corneal epithelial cells are GFP-labeled donor derived and characterized.**

A    Representative GFP-labeled blastomeres that contributed to the chimeric porcine RPE cells are shown with immunofluorescence staining. White arrows identify the RPE cells. Scale bars, 25 µm.

B    Representative GFP-labeled blastomeres that contributed to the chimeric porcine corneal epithelial cells. Almost all RPE cells and corneal epithelial cells of the chimera stained with anti-GFP antibody. Scale bars, 75 µm.

C    Representative immunofluorescence images showed the expression of an RPE marker (BEST) in GFP-labeled cells. White arrows identify the RPE cells. Scale bars, 50 µm.

D    Representative immunofluorescence images showed the expression of a corneal epithelial cell marker (K12) in GFP-labeled cells. WT, Bama WT piglet. Mut, Bama $MITF^{L247S/L247S}$ piglet. CH, chimeric piglet. Scale bars, 50 µm.

Source data are available online for this figure.

& Nakauchi, 2015). By using the anophthalmic pig model, our study clearly demonstrated that in E60 chimeric fetuses, allogenic-characterized RPE cells can be produced using blastocyst complementation. The pig RPE cells formed as early as E28, which potentially avoids ethical issues about producing human RPE cells using pig–human chimeras (Garry & Garry, 2016).

In the current study, the chimeric pig with regenerated eyes was derived from the somatic cloned blastocyst complementation and developed to full term. This was validated that SCNT technology remains a feasible strategy to enable limitless production of organ-disabled cloned embryos from cultured cells. However, the chimeric pig cloning efficiency is quite low in this study and needs to be improved for producing allogenic organs more efficiently. These low efficiencies might result for various reasons. The genetic background (breeds and individual pigs) of the donor cell lines greatly impact the SCNT efficiency in pigs (Li *et al*, 2013; Hua *et al*, 2016) and miniature pigs, resulting in lower cloning efficiencies (Zhao *et al*, 2009). In the current study, a Large White cell line and a Bama miniature pig cell line were used as donor and host cells, respectively. Accordingly, Matsunari *et al* used the Duroc × Berkshire hybrid cell line as a donor and the Large White/Landrace × Duroc hybrid cell line as a host or the Large White/Landrace × Duroc hybrid cell line was used both as a donor and a host. Interestingly, the chimerism efficiencies in the chimeric piglets were higher when the cells with same genetic background were used as both the donor and the host (30.8% with the Large White/Landrace × Duroc hybrid cell line as both donor and host vs. 16.7% with the Duroc × Berkshire hybrid cell line as the donor and the Large White/Landrace × Duroc hybrid cell line as the host (Matsunari *et al*, 2013). These results also demonstrate that different cell lines greatly impact the chimerism efficiency. The SCNT procedures in two groups, including oocyte maturation, micromanipulation, embryo activation, and embryo culture medium, were quite different; all those factors might affect the pregnancy rate and thus reflect the number of embryos used. Lastly, we complemented the host embryos at the 4- to 8-cell stage, while Matsunari *et al* complemented the host embryos at the morula stage. Since only a fraction of the 4- to 8-cell stage embryos can develop to morulas, this might contribute to the differences in the number of embryos used. Alternatively, although $MITF^{L247S/L247S}$ pigs show weak viability and extra care is required for growing mutant pigs to adulthood, we could still propagate $MITF^{L247S/L247S}$ pigs using natural mating to produce eye-disabled embryos at a lower cost than SCNT.

Strikingly, we observed intact eyes in the chimeric pig and their origin was determined to be derived from GFP-labeled allogenic blastomeres. The various components of the eyes derived from allogenic pluripotent cells were normal in their configuration. This is in accordance with earlier work in which allogenic pancreas was generated *in vivo* in an apancreatic cloned pig (Matsunari *et al*, 2013). So far, there are no characterized porcine ES and iPS cells available for producing chimera; therefore, we used SCNT as a source of porcine pluripotent stem cells for blastocyst complementation. Recently, human ESC or iPSC present the ability to generate post-implantation interspecies chimeric embryos, further extending the possibility of producing human PSC-derived organs in large animals (Wu *et al*, 2017; Yang *et al*, 2017).

The ethical issues associated with the generation of human organs by complementing pig embryos are multifaceted and mainly focus on the contribution of human PSCs to the brain and germline (Garry & Garry, 2016). Thus, the human PSCs may need to be genetically modified to impair their contribution to the developing animal brain or germline. Another solution may come from the observation that the formation of female germ cells from allogenic cells could be suppressed when a male host embryo was complemented with female donor cells (Matsunari *et al*, 2013). In the current study, most but not all of the generated RPE and cornea cells were derived from GFP-labeled allogenic cells, raising another issue of using allogenic organs for clinical transplantation. This phenomenon has appeared in the previous complementation studies (Kobayashi *et al*, 2010; Matsunari *et al*, 2013; Wu *et al*, 2017). However, Kobayashi *et al* (2010) reported that pancreatic islets were entirely derived from donor rat iPSCs in their study generating normally functioning rat pancreas in $Pdx1^{-/-}$ mice. The disparity between these studies could be a result of the different genes targeted for making the organ-defective model or the different species used in the experiments. In addition, a recent paper by the Nakauchi group described mouse islets, generated in rats, being transplanted into diabetic mice. The mouse islets contained substantial numbers of rat cells, but they engrafted and essentially cured drug-induced diabetes in mice without long-term use of immunosuppression (Yamaguchi *et al*, 2017). These data indicated that even PSC-derived organs, generated in a xenogeneic host, that are not 100% derived from donor cells can still provide a therapeutic benefit.

In this study, the results demonstrated that MITF, as a master regulator, is both necessary and sufficient to specify PSCs and direct them to differentiate to distinct lineages for the formation of eyes. The approach described here provides the basis for the potential future application of generating personalized human patient-specific organs or functional cells in large animals that can be subsequently used for transplantation or advanced therapies.

## Materials and Methods

### Chemical and reagents

Unless indicated differently, all of the chemicals obtained were purchased from Sigma, St. Louis, MO, USA.

### Animal care

All experiments involving pigs were approved by the Guidelines for the Care and Use Committee of Institute of Zoology, Chinese Academy of Sciences. Pigs were raised at the Beijing Farm Animal Research Center. Each pregnant surrogate was raised in one column access to water and food.

### Establishment and culture of donor fibroblasts

Pig embryonic fibroblast cells (PEFs) were isolated from a 35-day-old male $MITF^{L247S/L247S}$ Bama miniature pig and a 35-day-old female LW pig. The pCAG-EGFP plasmid was transfected into Bama male PEFs by electroporation as described previously (Ross *et al*, 2010). The RFP-labeled fibroblast cells were isolated from a 35-day-old male Bama miniature pig. They were cultured in Dulbecco's modified Eagle's medium (DMEM; Gibco, Grand Island, NY, USA) supplemented with 15% fetal bovine serum (FBS; HyClone, Logan, UT, USA) in 5% $CO_2$ in humidified air at 38.5°C. After 90% confluency, fibroblasts were frozen in FBS with 10% DMSO and stored in liquid nitrogen. Donor cells were induced into quiescence by being grown to confluence and trypsinized before use, and then resuspended in TALP-H with 15% FBS.

## Oocyte culture

Slaughter porcine ovaries were collected and transported to the laboratory within 2 h in a vacuum flask (30–35°C). Then, the follicles between 3 mm and 6 mm in diameter were aspirated using an 18-gauge needle attached to a 10-ml syringe. Cumulus–oocyte complexes (COCs) were rinsed three times in HEPES-buffered Tyrode's medium containing 0.01% PVA in an incubator at 37°C. After washing three times in IVM medium, a group of 70–80 COCs was placed into wells of four-well cell culture plates (Nunc, Roskilde, Denmark) containing 500 μl of *in vitro* maturation medium and 400 μl mineral oil per well and cultured for 42–44 h at 39°C and 5% $CO_2$ in air (100% humidity). Cumulus cells were removed by 0.1% hyaluronidase in HEPES-buffered Tyrode's medium containing 0.01% PVA for 5 min. The matured oocytes having an extruded first polar body (PB) were individually observed under stereoscopic microscopy (Nikon, Tokyo, Japan) for further use.

## Somatic cell nuclear transfer

After 42–44 h of maturation, the *in vitro*-matured oocytes were enucleated as recipient cytoplasts and fused with a single intact donor cell by using two direct pulses of 1.2 kV/cm for 30 μs (BTX Electro Cell Manipulator 200) in 0.3 M mannitol, 1.0 mM $CaCl_2$, 0.1 mM $MgCl_2$, and 0.5 mM Hepes (pH adjusted to 7.0–7.4). The reconstructed embryo was placed in 500 μl porcine zygote medium-3 (PZM3) containing 500 nM Scriptaid, a histone deacetylase inhibitor, and cultured for 14–16 h at 39°C in humidified 5% $CO_2$. Then, embryos were transferred to PZM3 medium for 4 days.

## Blastocyst complementation and embryo transfer

Blastocyst complementation and embryo transfer experiments were carried out as described previously (Matsunari *et al*, 2013). Pregnancy was diagnosed after 30 days and then was checked regularly at 4-week intervals by ultrasound examination. All of the piglets were delivered by natural birth.

## Genotyping for chimera

Blastocysts from the blastocyst complementation assay were transferred to the bottom of a 0.2-ml PCR tube under a stereomicroscope (Nikon), and each tube contained one blastocyst in 1 μl of PBS (pH 7.4). Single blastocyst genotyping was conducted according to the method described by Wang *et al* (2015). Genomic DNA was extracted from newborn piglets and fetuses using the TIANamp Genomic DNA Kit (Tiangen, China). DNA samples were analyzed using PCR with specific primers for the *MITF* gene and restriction fragment length polymorphism (RFLP) analysis. Eight microliters of PCR product was digested with *DraI* (New England Biolabs, USA). The full-length reaction products were 471 bp and were separated into 261 and 210 bp by 2% agarose gel electrophoresis in the presence of ethidium bromide solution and visualized with a UV transilluminator (UVP, Upland, CA, USA). Briefly, different bands were expected for genotyping. *MITF*[L247S/L247S] mutations were assessed by the loss of restriction enzyme sites harbored in the mutation region. PCR products were cut by *DraI*. Uncut bands were expected for mutant alleles. The sequences of the PCR primers are shown in Appendix Table S1.

## H&E and immunofluorescence staining

The eyes of pigs were fixed in 4% paraformaldehyde for 1 week, embedded in paraffin, sectioned, and stained. Sections stained with H&E were used to observe the structure of the eye. The samples were stained with MITF antibody (1:500; Abcam, Cambridge, MA), PAX6 antibody (1:50; Abcam, Cambridge, MA), RPE65 (1:500; Santa Cruz Biotechnology Inc., CA, USA), ZO-1 (1:500; Invitrogen, California, MA), K12 (1:100; Santa Cruz Biotechnology Inc., California, USA), and Bestrophin antibody (1:200; Abcam, Cambridge, MA) according to the manufacturer's protocol overnight at 4°C. The second antibodies used an Alexa Fluor 594 goat anti-mouse IgG (1:200) and an Alexa Fluor 488 goat anti-rabbit IgG (1:200) according to the manufacturer's protocol (ZSGB-Bio, Beijing, China) for 1 h at RT. After antibody treatments, sections were mounted in H33342 (Vector Laboratory Inc., Burlingame, CA, USA) for nuclear counterstaining and examined by confocal laser scanning microscopy using an LSM 5 PASCAL (ZEISS, Oberkochen, Germany).

## Culture of porcine RPE cells *in vitro*

Porcine RPE cells were isolated and cultured as previously described (Heller *et al*, 2015).

## Quantification of RPE markers

To investigate the abundance of mRNA in cultured RPE cells, total RNA was extracted from the samples using TRIzol reagent (Ambion by Life Technology, Waltham, USA) according to the manufacturer's protocol. All of the primer sequences are shown in Appendix Table S1.

## Western blotting

The RPE cells were trypsinized from 6-well flasks, frozen immediately in liquid nitrogen and stored at −80°C until use. The *ZO-1, RPE65,* and *Bestrophin* antibodies (1:1,000) were used to detect the expression of those proteins. *GAPDH* (1:3,000; CWBIO, Beijing, China) was used as a housekeeping gene to confirm equal sample loading.

## ELISA

VEGF and PEDF levels were measured in samples collected from the apical and basal sides of the same insert ($n = 3$). The VEGF and PEDF ELISA were performed according to kit instructions (Cloud-Clone, Houston, USA, and Elabscience, Bethesda, USA, respectively).

## PCR products mixing and purification

Each *MITF* gene PCR product from various tissues of NW-2 and NW-16 were mixed with same volume of 1× loading buffer (contained SYB green), and electrophoresis on 2% agarose gel was operated for quality checking. Samples showing bright main strip at 400 bp were chosen for further experiments. PCR products of all samples were mixed in equal density, and the mixture PCR products were purified with GeneJET Gel Extraction Kit (Thermo Scientific).

### The paper explained

**Problem**

Many people die of end-stage organ failure due to the shortage of human donor organs. Xenogeneic sources for transplantation are a promising alternative strategy for providing organs to treat certain diseases, such as age-related macular degeneration (AMD). However, regeneration of more complicated organs, such as eyes, in pigs with an empty "organ niche", has not yet been studied.

**Results**

Allogenic retinal pigmented epithelium cells (RPEs) were retrieved from an E44 and E60 chimeric fetus created using blastocyst complementation. Finally, whole allogenic eyes with normal structure were rescued from full-term chimeric piglets.

**Impact**

These results not only provide an important anophthalmic pig model toward generation of xenogeneic human eyes from patient-derived PSCs, but also provide novel insights into ocular organogenesis. This study adds more evidence to the feasibility of developing functional organs from xenogenic PSCs in large animals as a potential solution to the shortage of human donor organs worldwide.

The *MITF*-specific primer sequences were as follows: forward sequence 5′-CCACAGAGTCTGAAGCGAGA-3′ and reverse sequence 5′-CAGTTCTCCAGGGTAGGTTCC-3′.

### Library preparation and sequencing

The sequencing libraries were established by *MITF*-specific primer with different "Barcode" in the 5′-end of primers. The $MITF^{+/L247S}$ pig PCR product was used as positive control (WT allele/total allele = 50%). Then, the PCR products were generated in liver, lung, kidney, and eye of NW-2 fetus and in lung, kidney, testis, and eye of NW-16 piglet, respectively. Sequencing libraries with index codes were generated using NEB Next® Ultra DNA Library Prep Kit for Illumina (NEB, USA) according to manufacturer's recommendations. Quality of the library was assessed on the Qubit 2.0 Fluorometer (Thermo Scientific) and Agilent Bioanalyzer 2100 system. The library was sequenced on an Illumina HiSeq platform, and 250-bp paired-end reads were generated. The original sequence reads were split based on the index sequences, and then, the clean reads for each sample were mapped to pig build 10.2 reference sequence using Burrows-Wheeler Aligner (BWA) tool. Next, the number of reads containing WT allele and mutant allele were counted for assessing frequency of chimerism (WT allele/total allele).

### Statistics

All data were subjected to analysis by paired *t*-test, two-tailed using the GraphPad Prism software. Differences between treatment groups were considered significant at $P < 0.05$.

**Expanded View** for this article is available online.

### Acknowledgements

We are grateful to members of Yu laboratories from Zhejiang University for kindly providing ARPE-19 cells; thank Q. Meng from Institute of Zoology, Chinese Academy of Sciences for help on FACS sorting; thank S.W. Li and X.L. Zhu from Institute of Zoology, Chinese Academy of Sciences for help on confocal microscopy. This study was supported by the Strategic Priority Research Program of the Chinese Academy of Sciences (XDA16030101 and XDA16030300), the National Natural Science Foundation of China (81671274, 31272440, 31801031), and the National Transgenic Project of China (2016ZX08009003-006-007).

### Author contributions

QZ and JZ conceived the study and designed the experiments. HZ and JH performed the experiments. ZL conducted the blastocyst complementation and ET experiments. GQ and QZ provided help with SCNT and ET, respectively. NZ, TH, and RS helped with oocyte collection and ET. YZ, QH, and JY provided help with H&E staining. CC helped with the analysis of the next-generation sequencing data. JZ and HZ wrote the manuscript.

### Conflict of interest

The authors declare that they have no conflict of interest.

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
