## [Review Process File · EMBO Molecular Medicine]

Rescuing ocular development in an anophthalmic pig by blastocyst complementation

Hongyong Zhang, Jiaojiao Huang, Zechen Li, Guosong Qin, Nan Zhang, Tang Hai, Qianlong Hong, Qiantao Zheng, Ying Zhang, Ruigao Song, Jing Yao, Chunwei Cao, Jianguo Zhao, Qi Zhou

Review timeline:

Submission date:	11 January 2018
Editorial Decision:	22 February 2018
Revision received:	23 August 2018
Editorial Decision:	18 September 2018
Revision received:	24 September 2018
Editorial Decision:	01 October 2018
Revision received:	08 October 2018
Accepted:	11 October 2018

Editor: Céline Carret

Transaction Report:

1st Editorial Decision

22 February 2018

Thank you for the submission of your manuscript to EMBO Molecular Medicine. We have now heard back from the three referees whom we asked to evaluate your manuscript.

You will see from the set of comments pasted below that all referees find the study important and well executed. They all commend the amount of work provided and agree on the clinical relevance as a proof of concept strategy. We would like to draw your attention to the comments from referees 2 and 3 who request evidence of vision (and hearing) for the chimeras. We strongly would encourage you to perform these functional assays, along with providing more details of experimental settings, numbers and also show data that currently is alluded to but not reported.

We would therefore welcome the submission of a revised version within three months for further consideration and would like to encourage you to address all the criticisms raised as suggested to improve conclusiveness and clarity. Please note that EMBO Molecular Medicine strongly supports a single round of revision and that, as acceptance or rejection of the manuscript will depend on another round of review, your responses should be as complete as possible.

I look forward to receiving your revised manuscript.

***** Reviewer's comments *****

Referee #1 (Remarks for Author):

In their manuscript "Rescuing ocular development in an anophthalmic pig by blastocyst complementation" the author report the successful rescue of the eye histology and function in MITFL247S/L247 pig mutant by blastocysts complementation with allogeneic blastomeres obtained by somatic cell nuclear transfert (SCNT). This work adds to the burgeoning field of rescue of mutant animals by blastocyst complementation, in a species of interest for human health as pigs are considered as a reasonable option to grow human organs in animal by interspecies blastocyst complementation.

The authors first show in vitro allogeneic blastocysts complementation between WT or MITFL247S/L247 mutant (displaying "anophthalmia") Bama miniature pigs with large white GFP-positive blastomeres. Then, cloned embryos derived from MITFL247S/L247 male embryonic fibroblasts were used as the host and blastomeres of cloned morula derived from female LW embryonic fibroblasts were used as donors for complementation. A total of 3754 embryos were transferred to 16 surrogate sows, and 3 became pregnant from which 2 terminated during early pregnancy and one gave 4 fetuses at E44 by caesarean section. From these 4 fetuses, one was chimeric. The complemented fetus didn't display the anophthalmic phenotype displayed by the non-complemented fetuses. In another blastocyst complementation experiment (MITFL247S/L247 embryos injected with Bama WT blastomeres and pregnancy terminated at E60) RPE cells were obtained from a chimeric fetus and characterized as normal. Finally, in a last series of experiments, the authors complemented MITFL247S/L247 embryos with GFP-labeled Bama blastomeres (1671 reconstructed embryos, transferred into 7 surrogate sows, resulting in 3 pregnancies, two of which developed to term, resulting in 7 live-born piglets were obtained, one of which was chimeric). In this chimeric piglet, almost all the RPE cells and corneal epithelial cells of the chimeric piglet were GFP positive.

A considerable work was carried out for this manuscript, totalizing several thousand pig embryos reconstructed by blastomere complementation using cloned embryos, including several combinations of donor/recipient embryo (involving two different pig breeds, mutant versus WT, GFP positive or not) that were transferred to tens of surrogate sows. To my knowledge this is the first report of the proof of principle of repairing an eyes defect in a mutant pig by blastocyst complementation. It is also the first report of the repair of a in a MITF eye mutant by blastocyst complementation. Blastocyst complementation was already reported in pig in an apancreatic pig model, and the rescue of the eyen of a pax6 mutant was already reported in a mouse/mouse model (Wu et al Cell 2017) by complementing the embryos with iPS cells.

However, I have several concerns regarding this manuscript.

1/ The claim that the WT blastomeres rescued the eye of the MITFL247S/L247 fetuses can only be based on the observation that chimerism is higher in the eye than in other organs. Indeed, if the GFP+ WT genotype is dominant in the fetus (and in some fetuses only donor (GFP+ WT) cells were observed), then it is expected to have a WT phenotype. In this regard, the contribution of GFP+ WT cells to the NW-16 fetus lung and kidney seems to be high according to immunohistochemistry. Therefore, more precise strategies to assess the percentage of chimerism in the eye versus other organs should be carried out. The analysis of neighbor tissues (choroid, etc.) maybe of interest in this regard. In addition, the sequencing analysis protocol of the expression of MITF in the eye of piglet NW-16 (Figure S3) should be more detailed. Overall, the authors should display clear data indicating that the chimeric fetuses and piglets, and in particular the NW-16 piglet are bona fide chimeric and quantify the level of chimerism in different tissues. This point is key for the significance of the manuscript.

2/ The authors claim that there is no immune rejection. However, there is no proof of this.

3/ The fact that blastocyst complementation was already reported in a mouse/mouse model (Wu et al Cell 2017) by complementing the embryos with iPS cells should be specifically acknowledged.

4/ There is no data regarding experiment #2 (MITFL247S/L247 embryos injected with Bama WT blastomeres) to indicate the number of reconstructed embryos, surrogates sow etc.

5/ Page 12, the meaning of the sentence "In this study, we indicated that MITF, as master regulators, are both necessary and sufficient to specify PSCs and direct them to differentiate to distinct lineages" is not clear.

6/ It should be indicated in the discussion that the approach used in this manuscript cannot be applied to produce human/pig interspecies chimera for the production of human organ in pigs as the use of SNCT as a source of human cells for blastocyst complementation is unlikely due to debated ethical issues.

7/ The number of reconstructed embryos is more than 10 times higher than that used in the Matsunari et al. 2013 paper. The authors should discuss this point.

8/ In the abstract sentence "but also a disabled organ with highly specialized constituent tissues can be generated from exogenous PSCs when delivered to pig embryos with an empty organ niche" should be corrected and the term "PSC" replaced by "blastomere".

9/ Page 5 "The findings confirm that blastocyst complementation is feasible in vitro with PEFs. " : this sentence is not exact as the blastocyst complementation is feasible in vitro using blastomeres obtained from PEFs via SCNT, but not directly by injecting PEF into the blastocyst.

Likewise : "To generate allogenic chimeric pigs, we first explored the possibility of blastocyst complementation in vitro using pig embryonic fibroblast cells (PEFs)"

10/ "Alternatively, using an empty organ niche to produce functional xenogenic cells, such as RPEs, in the early stage of pregnancy for age-related disease might raise fewer ethical questions" The significance of this sentence is not clear. It may not be necessary to keep here.

11/ Syntax, terminology, and spelling checks should be conducted.

Other minor concerns:

Page 12 "In this study, we indicated that MITF" : rather we "demonstrate"

Figure 3G: contrary to what is indicated in the Figure legend, significant differences are missing

Legend of table S2 is incomplete

Referee #2 (Remarks for Author):

The manuscript provides some evidence for the feasibility of generating allogenic pig eye organ in an anophthalmic niche. The scale of the work is impressive. However, conceptually, and based on the previous papers by the same group, the novelty is not there. From a clinical point of view, the strategy using blastomeres as a donor for blastocyst complementation is absolutely impractical for clinical ocular disease.

The authors claim that the RPE cells derived from chimeras are functional, but this was not demonstrated. The authors show morphological similarities between the WT and chimera retina and demonstrate expression of retinal proteins, but functional assays like ROS phagocytosis assays or transplantation in an animal model of retinal disease were not performed. Furthermore, 1 chimera (NW-16) was born alive with eye of normal size and morphology so it would have been interesting to know whether or not NW-16 had vision.

Figure 2A-B demonstrates that NW-1 and NW-3 are not chimeras and are derived from host embryos (MITFL247S/L247S). However, they appear to have eyes morphologically similar to the WT embryo. Why is that?

Why weren't Immunofluorescence images of MITF reins included in Figure 2D as a negative control?

Is Figure 3G representative of only chimeric porcine RPE cells? If so, why weren't WT cells included in the analysis as a control?

Referee #3 (Remarks for Author):

In the manuscript entitled "Rescuing ocular development in an anophthalmic pig by blastocyst complementation", the authors describe their tour-de-force attempt to rescue ocular development in embryos homozygous for a microphthalmia-associated transcription factor (MITF) mutation by utilizing somatic cell nuclear transfer (SCNT) and blastocyst complementation techniques. These technically highly demanding experiments were well executed and provided results that could be potentially useful for future application of this approach to generate human eyes. There are several issues that need to be addressed before publication of this exciting paper.

Major comments:

According to the paper that originally described generation of the MITFL247S/L247S miniature pig which is a pig model of Waargenburg syndrome type 2A (Hai et al. Human Genet. 2017), these pigs are not anophthalmic, but microphthalmic and hypopigmented, due to homozygous mutation in the gene encoding microphthalmia-associated transcription factor (MITF). Figure 4 of this paper presents histologic images of a MITFL247S/L247S miniature pig eye. These show presence of retinal cells and choroid while retinal pigmented epithelium (RPE) is absent. The authors need to explain why they consider these animals to model anophthalmia rather than microphthalmia.

The photomicrograph of retinal structures in the mutant pig shown in Figure 4B of the current paper differs from one shown in the paper by Hai et al. (*ibid.*). If the retina in MITFL247S/L247S pigs is totally absent, the retinal cells in complemented pig eyes should be 100% donor-derived. Figure 4D is not clear enough to show how donor and host retinal cells are organized. This point is important when considering application of this method to generate donor retinal cells. The authors mention that not all of the generated RPE and cornea cells were derived from donor cells. This could be due to the presence of remaining host cells in MITFL247S/L247S pigs, a point that must be addressed in greater detail.

In the Discussion, the authors should cite a recent paper by the Nakauchi group describing transplantation into diabetic mice of mouse islets generated in rats. Those islets contained substantial numbers of rat cells, but they engrafted and essentially cured drug-induced diabetes without long-term use of immunosuppression (Yamaguchi et al. Nature 2017).

The MITFL247S/L247S mutation is known to cause hearing defects. Did blastocyst complementation rescue this phenotype as well? Functional evidence for the pigs' vision and hearing capacity after birth would add value to this paper.

Histological analysis that delineates the extent of complementing-cell derived, or GFP-positive elements in the host ocular structure should be shown.

Minor comments:

The quality of the photos is in general not high enough. Figure 1 can go into a supplement and give more space for images in Figure 4.

Did the extent of donor chimerism affect ocular complementation?
Were there any chimeras without successful ocular complementation?

The style is generally excellent. However, some phrases or sentences seem to have been added after the manuscript left the hands of a native speaker ("...cell replacement therapies, However, the former was still need to be optimized..."). Please give any revised version a final tidying before submission.

Reviewer's comments

Referee #1 (Remarks for Author):

In their manuscript "Rescuing ocular development in an anophthalmic pig by blastocyst complementation" the author report the successful rescue of the eye histology and function in MITF^{L247S/L247S} pig mutant by blastocysts complementation with allogeneic blastomeres obtained by somatic cell nuclear transfer (SCNT). This work adds to the burgeoning field of rescue of mutant animals by blastocyst complementation, in a species of interest for human health as pigs are considered as a reasonable option to grow human organs in animal by interspecies blastocyst complementation.

The authors first show in vitro allogeneic blastocysts complementation between WT or MITF^{L247S/L247S} mutant (displaying "anophthalmia") Bama miniature pigs with large white GFP-positive blastomeres. Then, cloned embryos derived from MITF^{L247S/L247S} male embryonic fibroblasts were used as the host and blastomeres of cloned morula derived from female LW embryonic fibroblasts were used as donors for complementation. A total of 3754 embryos were transferred to 16 surrogate sows, and 3 became pregnant from which 2 terminated during early pregnancy and one gave 4 foetuses at E44 by caesarean section. From these 4 foetuses, one was chimeric. The complemented foetus didn't display the anophthalmic phenotype displayed by the non-complemented foetuses. In another blastocyst complementation experiment (MITF^{L247S/L247S} embryos injected with Bama WT blastomeres and pregnancy terminated at E60)

RPE cells were obtained from a chimeric foetus and characterized as normal. Finally, in a last series of experiments, the authors complemented MITF^{L247S/L247S} embryos with GFP-labelled Bama blastomeres (1671 reconstructed embryos, transferred into 7 surrogate sows, resulting in 3 pregnancies, two of which developed to term, resulting in 7 live-born piglets were obtained, one of which was chimeric). In this chimeric piglet, almost all the RPE cells and corneal epithelial cells of the chimeric piglet were GFP positive.

A considerable work was carried out for this manuscript, totalizing several thousand pig embryos reconstructed by blastomere complementation using cloned embryos, including several combinations of donor/recipient embryo (involving two different pig breeds, mutant versus WT, GFP positive or not) that were transferred to tens of surrogate sows. To my knowledge this is the first report of the proof of principle of repairing an eyes defect in a mutant pig by blastocyst complementation. It is also the first report of the repair of a in a MITF eye mutant by blastocyst complementation. Blastocyst complementation was already reported in pig in an apancreatic pig model, and the rescue of the eye of a pax6 mutant was already reported in a mouse/mouse model (Wu et al Cell 2017) by complementing the embryos with iPS cells.

However, I have several concerns regarding this manuscript.

1/ The claim that the WT blastomeres rescued the eye of the MITF^{L247S/L247S} foetuses can only be based on the observation that chimerism is higher in the eye than in other organs. Indeed, if the GFP+ WT genotype is dominant in the foetus (and in some foetuses only donor (GFP+ WT) cells were observed), then it is expected to have a WT phenotype. In this regard, the contribution of GFP+ WT cells to the NW-16 foetus lung and kidney seems to be high according to immunohistochemistry. Therefore, more precise strategies to assess the percentage of chimerism in the eye versus other organs should be carried out. The analysis of neighbour tissues (choroid, etc.) may be of interest in this regard. In addition, the sequencing analysis protocol of the expression of MITF in the eye of piglet NW-16 (Figure S3) should be more detailed. Overall, the authors should display clear data indicating that the chimeric foetuses and piglets, and in particular the NW-16 piglet are bona fide chimeric and quantify the level of chimerism in different tissues. This point is key for the significance of the manuscript.

Response: Thanks for the suggestions on the evaluation of chimerism contributions.

As suggested by the reviewer, we further evaluated the chimerism contribution by using next generation sequencing in liver, lung, kidney and eye of E44 fetuses and in lung, kidney, testis and eye of NW-16 piglets, respectively. Results showed the chimeric contributions in 44-day fetus was on average ~27% and no differences in the tissues were determined (liver was 27.85%, lung was 28.06%, kidney was 27.12% and eye was 27.14%, respectively). Higher levels of chimerism in the eyes were observed in the NW-16 piglets, which is consistent with the results from Matsunari et al. (lung was 66.76%, kidney was 65.37%, testis was 41.94% and eye was 85.76%). This data has been added in new Figure EV4 in the revised manuscript. Because we choose a more precise method to

detect the chimerism contributions, we abandoned the previous data from the old version's Figure S3.

Figures for Referees not shown

2/ The authors claim that there is no immune rejection. However, there is no proof of this.

Response: Thanks for the comment. Although we determined the successful chimerism in the full term piglet (NW-16) as well as the E44 fetus, we don't have data to support the claim of no immune rejection. As the reviewer suggested, to be more accurate, we deleted this point because we did not examine the immune system. The changes are in the revised version on line 188, line 235 and line 263.

3/ The fact that blastocyst complementation was already reported in a mouse/mouse model (Wu et al. Cell 2017) by complementing the embryos with iPS cells should be specifically acknowledged.

Response: Thanks for the suggestion. The suggested paper has been added and acknowledged. Please see changes in red font in the revision manuscript line 56-57.

4/ There is no data regarding experiment #2 (MITF^{L247S/L247S} embryos injected with Bama WT blastomeres) to indicate the number of reconstructed embryos, surrogates sow etc.

Response: A total of 821 embryos were transferred to 3 surrogate sows, and 1 sow became pregnant. The data has been added in the revised version line 170-171.

5/ Page 12, the meaning of the sentence "In this study, we indicated that MITF, as master regulators, are both necessary and sufficient to specify PSCs and direct them to differentiate to distinct lineages" is not clear.

Response: Based on the reviewer's suggestion, we change the sentence to "In this study, the results demonstrated that MITF, as a master regulator, is both necessary and sufficient to specify PSCs and direct them to differentiate to distinct lineages for the formation of eyes". The changes are in red font on line 290-292.

6/ It should be indicated in the discussion that the approach used in this manuscript cannot be applied to produce human/pig interspecies chimera for the production of human organ in pigs as the use of SNCT as a source of human cells for blastocyst complementation is unlikely due to debated ethical issues.

Response: So far, there are no characterized porcine ES or iPS cells available for producing chimera. This is the reason we use SCNT derived blastomeres as the donor cells in the current study. However, human iPS cells are available (Jansch et al, 2018; Mura et al, 2018; Park et al, 2008) and can be used for blastocyst complementation. From the current study, we provide proof-of-concept that a more sophisticated organ, such as eyes, can be generated when organogenesis-disabled embryos are complemented with allogenic blastomeres. We will explore the possibility of using this pig model for producing interspecies chimeras to regenerating human RPEs by injecting specific human patient iPSCs instead of human blastomeres. We discuss this point in the revised version on line 265-267.

7/ The number of reconstructed embryos is more than 10 times higher than that used in the Matsunari et al. 2013 paper. The authors should discuss this point.

Response: Thanks for the comment. Actually, we noticed the low efficiencies of producing chimeras in this study with current cell lines. This might result from:

1. The genetic background (breeds and individuals) donor cell line greatly impacts the SCNT efficiency in pigs and (Hua et al, 2016; Li et al, 2013) miniature pigs, which have lower cloning efficiency (Zhao et al, 2009). In the current study, Bama miniature pig cell lines and Large White cell lines were used as donor and host cells, respectively. Accordingly, Matsunari et al. used the Duroc×Berkshire hybrid cell line as a donor, the Large White/Landrace×Duroc hybrid cell line as a host, or the Large White/Landrace×Duroc hybrid cell line was used as both a donor and host. Interestingly, the chimerism efficiencies in the chimeric piglets were higher when the

cells with the same genetic background were used as both the donor and host (30.8% with the Large White/Landrace×Duroc hybrid cell line as both the donor and host Vs. 16.7% with the Duroc×Berkshire hybrid cell line as the donor and the Large White/Landrace×Duroc hybrid cell line as the host (Matsunari et al, 2013). These results demonstrated that the different cell lines greatly impact chimerism efficiency.

2. The SCNT procedure in the two groups, including oocyte maturation, micromanipulation, embryo activation, and embryo culture medium, are different. All those factors might affect the pregnancy rate and thus reflect the number of embryos used.
3. Last, we complemented the host embryos at the 4-8 cell stage; however, Matsunari et al. complemented the host embryos at the morula stage. Since only a fraction of the 4-8 cell stage embryos can develop to morulas, this might contribute to the differences in the number of embryos used.

This point has been discussed in the revised manuscript on line 238-257.

8/ In the abstract sentence "but also a disabled organ with highly specialized constituent tissues can be generated from exogenous PSCs when delivered to pig embryos with an empty organ niche" should be corrected and the term "PSC" replaced by "blastomere".

Response: Corrected. See the changes in red font on line 36.

9/ Page 5 "The findings confirm that blastocyst complementation is feasible in vitro with PEFs." : this sentence is not exact as the blastocyst complementation is feasible in vitro using blastomeres obtained from PEFs via SCNT, but not directly by injecting PEF into the blastocyst. Likewise: "To generate allogenic chimeric pigs, we first explored the possibility of blastocyst complementation in vitro using pig embryonic fibroblast cells (PEFs)"

Response: Thanks for the good suggestion.

Based on the reviewer's suggestion, we change the first sentence to: "The findings confirm that blastocyst complementation is feasible *in vitro* with blastomeres derived from PEFs" and can be seen in red font on line 124. We changes the second sentence to: "To generate allogenic chimeric pigs, we first explored the possibility of blastocyst complementation in vitro using embryos derived from pig embryonic fibroblast cells (PEFs)" on line 103 in red font.

10/ "Alternatively, using an empty organ niche to produce functional xenogenic cells, such as RPEs, in the early stage of pregnancy for age-related disease might raise fewer ethical questions" The significance of this sentence is not clear. It may not be necessary to keep here.

Response: As suggested, this sentence has been deleted in the revised version.

11/ Syntax, terminology, and spelling checks should be conducted.

Response: Thanks for the suggestion.

We went through the whole manuscript carefully and made the necessary corrections.

Other minor concerns:

Page 12 "In this study, we indicated that MITF": rather we "demonstrate"

Response: Corrected in red font on line 290.

Figure 3G: contrary to what is indicated in the Figure legend, significant differences are missing

Response: we revised the figure legend on line 633. There is no significant difference.

Legend of table S2 is incomplete

Response: We made the correction.

Referee #2 (Remarks for Author):

The manuscript provides some evidence for the feasibility of generating allogenic pig eye organ in an anophthalmic niche. The scale of the work is impressive. However, conceptually, and based on

the previous papers by the same group, the novelty is not there. From a clinical point of view, the strategy using blastomeres as a donor for blastocyst complementation is absolutely impractical for clinical ocular disease.

Response: Thanks for the comments.

We do agree that using blastomeres as a donor for blastocyst complementation is impractical for clinical human ocular disease. So far, there are no characterized porcine ES and iPSC cells available for producing chimeras. This is the reason we use SCNT derived blastomeres as the donor cells in the current study. However, human iPSC cells are available (Jansch et al, 2018; Mura et al, 2018; Park et al, 2008) and can be used for blastocyst complementation. In the current study, we provide proof-of-concept that a more sophisticated organ, such as eyes, can be generated when organogenesis-disabled embryos are complemented with allogenic blastomeres. We will explore the possibility of using this pig model for producing interspecies chimeric to regenerate human RPEs by injecting specific human patient iPSCs instead of human blastomeres. We discuss this point in the revised version on line 265-267.

The authors claim that the RPE cells derived from chimeras are functional, but this was not demonstrated. The authors show morphological similarities between the WT and chimera retina and demonstrate expression of retinal proteins, but functional assays like ROS phagocytosis assays or transplantation in an animal model of retinal disease were not performed. Furthermore, 1 chimera (NW-16) was born alive with eye of normal size and morphology so it would have been interesting to know whether or not NW-16 had vision.

Response: Thanks for these good comments.

1. As matter of fact, we did not perform functional evaluation, but only determined VEGF and PEDF levels in the RPEs by ELISA assay. RPEs secreted VEGF and PEDF to maintain the choroidal blood supply (Maminishkis et al, 2006) and thus, VEGF and PEDF could be viewed as a specific marker to define RPE cells (Maminishkis et al, 2006). For more accuracy, we modified “functional RPEs” to “characterized RPEs”.
2. The NW-16 piglet died before we had the chance to examine its vision. However, the piglet could find the nipple of the surrogate sow for milk, which makes us believe that his vision was fully restored. Furthermore, the morphology and structure of the eyes of NW-16 were quite normal when compared with the WT piglet (Figure 4C and D, revised version). Confirming the vision of chimera is very important, we tried our hardest to obtain full term chimeric piglet for vision test. A total of 14 surrogate sows received ~200 embryos each, resulting in 5 pregnancies which gave birth of 11 piglets. Of the 11 piglets, 9 piglets derived from MITF^{L247S/L247S} cells, 2 fetuses derived from GFP-labeled Bama male PEFs and no chimeras were obtained. This was very frustrated results. Unluckily, we did not obtain the full-term chimeric piglets. Considering the long gestation of large animals and time consuming for somatic cell nuclear transfer, we couldn't make more embryo transfers and are pleading the reviewer to consider this manuscript under the circumstances.

Figures for Referees not shown

Figure 2A-B demonstrates that NW-1 and NW-3 are not chimeras and are derived from host embryos (MITF^{L247S/L247S}). However, they appear to have eyes morphologically similar to the WT embryo. Why is that?

Response: Thanks for the comments.

Although the appearance of NW-1 and NW-3 looks like WT embryos, we confirmed that NW-1 and NW-3 had abnormal retina structure as determined by hematoxylin and eosin (H&E) staining. H&E staining showed that the RPE cells in NW-1 and NW-3 were hypopigmented and disorganized (black arrowhead), and the iris and choroid were abnormal (black square frame). Some cells were observed that were “RPE-like” but they were not pigmented. These results were consistent with our previous research (Hai et al, 2017). Using RFLP assays, genotyping results also confirmed that NW-1 and NW-3 were totally derived from MITF^{L247S/L247S} mutant cell lines. Taken together, we presumed that MITF^{L247S/L247S} pigs lose RPE cells gradually and the retina disappears in the neonate.

Figures for Referees not shown

Why weren't Immunofluorescence images of MITF retinas included in Figure 2D as a negative control?

Response: Thanks for the good suggestion.

A negative control of immunofluorescence of the *MITF*^{L247S/L247S} retina has been included in Figure 2B in the revised manuscript. In the results, positive staining of *MITF*, *Pax6* and *Bestrophin* was observed in the regenerated RPE in the NW-2 fetus, suggesting functional RPE in NW-2. Meanwhile, in the mutant NW-1 fetus, we found that the subcellular distribution of *Pax6* and *MITF* was changed and showed negative expression of *Bestrophin*. We revised the description of Figure 2B in the main text (Line 153-156), as the reviewer suggested.

Figures for Referees not shown

Is Figure 3G representative of only chimeric porcine RPE cells? If so, why weren't WT cells included in the analysis as a control?

Response: Thanks for the good suggestion.

We have added the WT control in Figure 3G to the revised manuscript. The level of VEGF and PEDF are similar between WT and CH cells. Assay results showed trends that VEGF was secreted more into the basal bath (n = 3), whereas PEDF was secreted more into the apical bath (n = 3), but overall there was no significant difference (Fig. 3G). In the revised manuscript, Figure 3G (line 183) and figure legend (line 630) include those changes.

Figures for Referees not shown

Referee #3 (Remarks for Author):

In the manuscript entitled "Rescuing ocular development in an anophthalmic pig by blastocyst complementation", the authors describe their tour-de-force attempt to rescue ocular development in embryos homozygous for a microphthalmia-associated transcription factor (MITF) mutation by utilizing somatic cell nuclear transfer (SCNT) and blastocyst complementation techniques. These technically highly demanding experiments were well executed and provided results that could be potentially useful for future application of this approach to generate human eyes. There are several issues that need to be addressed before publication of this exciting paper.

Major comments:

According to the paper that originally described generation of the *MITF*^{L247S/L247S} miniature pig which is a pig model of Waargenburg syndrome type 2A (Hai et al. Human Genet. 2017), these pigs are not anophthalmic, but microphthalmic and hypopigmented, due to homozygous mutation in the gene encoding microphthalmia-associated transcription factor (MITF). Figure 4 of this paper presents histologic images of a *MITF*^{L247S/L247S} miniature pig eye. These show presence of retinal cells and choroid while retinal pigmented epithelium (RPE) is absent. The authors need to explain why they consider these animals to model anophthalmia rather than microphthalmia.

Response: Thanks for the suggestions.

Actually, microphthalmia was named after the *MITF* mutation in the mice. In mice, mutations in the *MITF* gene resulted in hearing loss phenotypes of recessive or semidominant inheritance patterns. However, in human and pigs, mutations in the *MITF* gene resulted in hearing loss phenotypes of a dominant inheritance pattern (Tassabehji et al, 1995). Recently, a paper reported that homologous mutations in the *MITF* gene cause phenotypes such as Coloboma, Osteopetrosis, Microphthalmia, Macrocephaly, Albinism, and Deafness in humans (George et al, 2016). In the Hai et al. 2017 paper, we demonstrated that hearing loss and hypopigmentation in skin, hair, and iris occurs in

heterozygous animals. Besides hypopigmentation and bilateral HL, the homozygous mutant pig ($MITF^{L247S/L247S}$) and CRISPR/Cas9-mediated *MITF* bi-allelic knockout pigs both exhibited more severe abnormalities in eyes rather than microphthalmia, which we called anophthalmia (See Figure below). H&E staining showed the structure of the eye was disorganization in newborn $MITF^{L247S/L247S}$ piglets (new figure 4D). Although Figure 4D showed that retinal cells were not completely formed (only analogous to ganglion cell layer) and showed an abnormal choroid in the $MITF^{L247S/L247S}$ pig, the abnormal eye phenotype was quite similar between humans and pigs. Considering these results, the abnormal eye development in $MITF^{L247S/L247S}$ pigs is a more suitable model for anophthalmia than microphthalmia.

Figures for Referees not shown

Response: Thanks for the suggestions.

The retinal structures shown in Figure 4B (old version) of the current paper was from a newborn mutant pig. However, the retinal structure of a 44-day mutant pig was characterized in the paper by Hai et al. We assume that this is the major reason for differences in the photomicrograph of retinal structures in the two mutant pigs.

Regeneration of disabled-organs has been studied in mouse-rat chimera, such as adult mouse-rat chimeras with $81.9\% \pm 3.4\%$ rat-derived cells in the pancreas (Kobayashi et al, 2010). Consistent with the research in mouse-rat chimeras, high levels of chimeric contributions to the eye were observed in the NW-16 piglet (85.76%) but not in those that were 100% donor-derived. One of the explanations for this phenotype is that the WT RPE cells might provide a niche for the mutant cells to develop into normal RPE cells. Furthermore, as the reviewer mentioned below, mouse islets that contained substantial numbers of rat cells prepared from mouse-rat chimeric pancreas were transplanted into diabetic mouse models and the transplanted islets successfully normalized and maintained host blood glucose levels for over 370 days (Yamaguchi et al, 2017). These data indicated that even the PSC-derived islets, generated in a xenogeneic host, were not 100% derived from donor cells, yet still provides therapeutic potential.

In Figure 4D (old version), the data showed that most of the retina cells were *GFP* positive in the chimeric piglet, which indicated they were derived from donor cells. As suggested by the reviewer, we have changed Figure 4D (old version) to Figure 5A in the revised manuscript to provide a better layout.

Although not all of the generated RPE cells were derived from donor cells, we can isolate the donor-derived RPE cells by specific labeling.

In the Discussion, the authors should cite a recent paper by the Nakauchi group describing transplantation into diabetic mice of mouse islets generated in rats. Those islets contained substantial numbers of rat cells, but they engrafted and essentially cured drug-induced diabetes without long-term use of immunosuppression (Yamaguchi et al. Nature 2017).

Response: Based on the reviewer's suggestion, we have cited the paper and discussed this viewpoint in our revised manuscript (Line 284-289).

The $MITF^{L247S/L247S}$ mutation is known to cause hearing defects. Did blastocyst complementation rescue this phenotype as well? Functional evidence for the pigs' vision and hearing capacity after birth would add value to this paper.

Response:

As suggested by the reviewer, we characterized the cochlea structure using celloidin embedding and H&E staining in 60-day WT, chimeric and $MITF^{L247S/L247S}$ fetuses. However, the results showed the cochlea structure was no difference between WT and $MITF^{L247S/L247S}$ fetuses. Thus, we are unable to determine whether or not the hearing capacity could be rescued in the 60-day fetus. NW-16 died before we were able to characterize its vision and hearing capacity. Confirming the vision of chimera is very important, we tried our hardest to obtain full term chimeric piglet for vision test. A total of 14 surrogate sows received ~200 embryos each, resulting in 5 pregnancies which gave birth of 11 piglets. Of the 11 piglets, 9 piglets derived from $MITF^{L247S/L247S}$ cells, 2 fetuses derived from GFP-labeled Bama male PEFs and no

chimeras were obtained. This was very frustrated results. Unluckily, we did not obtain the full-term chimeric piglets. Considering the long gestation of large animals and time consuming for somatic cell nuclear transfer, we couldn't make more embryo transfers and are pleading the reviewer to consider this manuscript under the circumstances.

Figures for Referees not shown

Histological analysis that delineates the extent of complementing-cell derived, or GFP-positive elements in the host ocular structure should be shown.

Response: Thanks for this suggestion.

Actually, we have detected the extent of GFP-positive elements in the host ocular structure, including retina and cornea in Figure 5A and 5B. In addition, next generation sequencing showed that the chimeric efficiency was 85.76% in the eye of NW-16.

Figures for Referees not shown

Minor comments:

The quality of the photos is in general not high enough. Figure 1 can go into a supplement and give more space for images in Figure 4.

Response: We appreciate the constructive suggestions from the reviewer, and we modified the Figures accordingly. We reorganized the new Figure 1 and Figure 2 entitled "Generation of E44 chimeric porcine fetus *in vivo* by complementation of *MITF*^{L247S/L247S} embryos with donor blastomeres derived from LW PEFs." and "Allogenic contribution and rescue RPEs in the E44 chimeric fetus", respectively. Specifically, the previous Figure 2 panel A and B were renamed as Figure 1 A and B, and the previous Figure 2 panel C and D were renamed Figure 2 panel A and B. To improve the quality of the new Figure, we moved the previous Figure 1 (old version) into the Supplemental data, now called Figure EV1 and Figure EV2. Specifically, the previous Figure 1 panel A, B, C, and D were renamed Figure EV1 panel C and D and Figure EV2 panel A and C. We added the VEGF and PEDF secreted by WT RPF cells, which were used as control, in the new panel G (prior versions of panel G). As the reviewer suggested, we also moved panel D, E, F and G of the prior Figure 4 into the new Figure 5 as panel A, B, C and D, respectively. We moved the panel of Figure S3F and G (the old versions) into the full text as the new Figure 4B. In addition, the result text and relevant figure legends have been updated accordingly.

Did the extent of donor chimerism affect ocular complementation? Were there any chimeras without successful ocular complementation?

Response: In our research, we did not find chimeras without successful ocular complementation, and the chimeric contributions were observed to be from ~27% (in 44 day embryo) to ~85.76% (in NW-16 piglet). It assumed that about ~27% chimeric contribution could rescue ocular development, and ocular rescue was identified in all surviving chimeric pigs. Furthermore, in the Kobayashi et al. paper, the lowest contribution (~5.35%) observed in the rat-mouse chimera still exhibited successful pancreas complementation (Kobayashi et al, 2010).

The style is generally excellent. However, some phrases or sentences seem to have been added after the manuscript left the hands of a native speaker ("...cell replacement therapies, however, the former was still need to be optimized..."). Please give any revised version a final tidying before submission.

Response: Thanks for the comments. The manuscript was again polished by a native speaker.

George A, Zand DJ, Hufnagel RB, Sharma R, Sergeev YV, Legare JM, Rice GM, Scott Schwoerer JA, Rius M, Tetri L et al (2016) Biallelic Mutations in *MITF* Cause Coloboma, Osteopetrosis, Microphthalmia, Macrocephaly, Albinism, and Deafness. American journal of human genetics 99: 1388-1394

Hai T, Guo W, Yao J, Cao C, Luo A, Qi M, Wang X, Wang X, Huang J, Zhang Y et al (2017)

Creation of miniature pig model of human Waardenburg syndrome type 2A by ENU mutagenesis. *Hum Genet* 136: 1463-1475

Hua Z, Xu G, Liu X, Bi Y, Xiao H, Hua W, Li L, Zhang L, Ren H, Zheng X (2016) Impact of different sources of donor cells upon the nuclear transfer efficiency in Chinese indigenous Meishan pig. *Polish journal of veterinary sciences* 19: 205-212

Jansch C, Gunther K, Waider J, Ziegler GC, Forero A, Kollert S, Svirin E, Puhlinger D, Kwok CK, Ullmann R et al (2018) Generation of a human induced pluripotent stem cell (iPSC) line from a 51-year-old female with attention-deficit/hyperactivity disorder (ADHD) carrying a duplication of SLC2A3. *Stem cell research* 28: 136-140

Kobayashi T, Yamaguchi T, Hamanaka S, Kato-Itoh M, Yamazaki Y, Ibata M, Sato H, Lee YS, Usui J, Knisely AS et al (2010) Generation of rat pancreas in mouse by interspecific blastocyst injection of pluripotent stem cells. *Cell* 142: 787-799

Li Z, Shi J, Liu D, Zhou R, Zeng H, Zhou X, Mai R, Zeng S, Luo L, Yu W et al (2013) Effects of donor fibroblast cell type and transferred cloned embryo number on the efficiency of pig cloning. *Cell Reprogram* 15: 35-42

Maminishkis A, Chen S, Jalickee S, Banzon T, Shi G, Wang FE, Ehalt T, Hammer JA, Miller SS (2006) Confluent monolayers of cultured human fetal retinal pigment epithelium exhibit morphology and physiology of native tissue. *Invest Ophthalmol Vis Sci* 47: 3612-3624

Matsunari H, Nagashima H, Watanabe M, Umeyama K, Nakano K, Nagaya M, Kobayashi T, Yamaguchi T, Sumazaki R, Herzenberg LA et al (2013) Blastocyst complementation generates exogenic pancreas in vivo in apancreatic cloned pigs. *Proceedings of the National Academy of Sciences of the United States of America* 110: 4557-4562

Mura M, Lee YK, Ginevrino M, Zappatore R, Pisano F, Boni M, Dagradi F, Crotti L, Valente EM, Schwartz PJ et al (2018) Generation of the human induced pluripotent stem cell (hiPSC) line PSMi002-A from a patient affected by the Jervell and Lange-Nielsen syndrome and carrier of two compound heterozygous mutations on the KCNQ1 gene. *Stem cell research* 29: 157-161

Park IH, Lerou PH, Zhao R, Huo H, Daley GQ (2008) Generation of human-induced pluripotent stem cells. *Nat Protoc* 3: 1180-1186

Tassabehji M, Newton VE, Liu XZ, Brady A, Donnai D, Krajewska-Walasek M, Murday V, Norman A, Obersztyn E, Reardon W et al (1995) The mutational spectrum in Waardenburg syndrome. *Hum Mol Genet* 4: 2131-2137

Yamaguchi T, Sato H, Kato-Itoh M, Goto T, Hara H, Sanbo M, Mizuno N, Kobayashi T, Yanagida A, Umino A et al (2017) Interspecies organogenesis generates autologous functional islets. *Nature* 542: 191-196

Zhao J, Ross JW, Hao Y, Spate LD, Walters EM, Samuel MS, Rieke A, Murphy CN, Prather RS (2009) Significant improvement in cloning efficiency of an inbred miniature pig by histone deacetylase inhibitor treatment after somatic cell nuclear transfer. *Biology of reproduction* 81: 525-530

2nd Editorial Decision

18 September 2018

Thank you for the submission of your revised manuscript to EMBO Molecular Medicine. We have now received the enclosed reports from the referees that were asked to re-assess it. As you will see the reviewers are now globally supportive and I am pleased to inform you that we will be able to accept your manuscript pending additional final editorial amendments.

Corresponding Author Name: Jianguo Zhao, Qi Zhou

Manuscript Number: EMM-2018-08861